# Viral Suppression and HIV Drug Resistance Among Patients on Second-Line Antiretroviral Therapy in Selected Health Facility in Ethiopia

**DOI:** 10.3390/v17020206

**Published:** 2025-01-31

**Authors:** Kidist Zealiyas, Atsbeha Gebreegziabxier, Yimam Getaneh, Eleni Kidane, Belete Woldesemayat, Ajanaw Yizengaw, Gadisa Gutema, Sisay Adane, Mengistu Yimer, Amelework Yilma, Sisay Tadele, Sviataslau Sasinovich, Patrik Medstrand, Dawit Assefa Arimide

**Affiliations:** 1Infectious Diseases Research Directorate, Ethiopian Public Health Institute, Addis Ababa 1242, Ethiopia; kzealiyas@gmail.com (K.Z.); atsbehag@gmail.com (A.G.); yimamgetaneh@gmail.com (Y.G.); elikid2003@gmail.com (E.K.); beleteweldesemeyat@gmail.com (B.W.); ajanawy19@gmail.com (A.Y.); gadissagutema@gmail.com (G.G.); sisay0229@gmail.com (S.A.); mengeyimer12@gmail.com (M.Y.); sameleworkyilema@yahoo.com (A.Y.); sisay75t@gmail.com (S.T.); 2Department of Translational Medicine, Lund University, 22185 Lund, Sweden; sviataslau.sasinovich@med.lu.se (S.S.); patrik.medstrand@med.lu.se (P.M.)

**Keywords:** antiretroviral therapy, HIV-1 drug resistance, viral suppression, second-line regimen, adherence, Ethiopia

## Abstract

HIV drug resistance (HIVDR) presents a significant challenge to antiretroviral therapy (ART) success, particularly in resource-limited settings like Ethiopia. This cross-sectional study investigated viral suppression rates and resistance patterns among patients on second-line ART across 28 Ethiopian health facilities. Blood samples collected from 586 participants were analyzed to measure CD4 count and viral load and assess HIVDR in patients experiencing virological failure (VF) (viral load ≥ 1000 copies/mL). Demographic and clinical data were analyzed using logistic regression to identify factors associated with VF. Results showed that 13.82% of participants experienced VF, with 67.57% of genotyped samples exhibiting at least one drug resistance mutation. Resistance to nucleoside reverse transcriptase inhibitors (NRTIs), non-nucleoside reverse transcriptase inhibitors (NNRTIs), and protease inhibitors (PIs) was detected in 48.64%, 64.86%, and 18.92% of cases, respectively. Dual-class resistance was identified in 48.64% of patients, while triple-class resistance was detected in 18.92%. VF was more likely among students and those with CD4 counts below 200 cells/mm³, but less likely in patients on second-line treatment for 12 months or more. Our findings highlight a substantial HIVDR burden among patients on second-line ART with VF, emphasizing the need for comprehensive HIV care, including adherence support, regular viral load monitoring, and HIVDR testing.

## 1. Introduction

Antiretroviral therapy (ART) roll-out has significantly improved the prognosis of HIV-infected individuals, transforming HIV from a fatal disease into a manageable chronic condition and potentially preventable condition [1,2]. However, the effectiveness of ART is increasingly threatened by the emergence and transmission of HIV drug resistance (HIVDR) [3,4,5]. This challenge is particularly significant in resource-limited settings where routine viral load (VL) and HIVDR monitoring are not consistently implemented due to financial, infrastructural, and logistical constraints. The lack of regular VL testing can result in undetected treatment failure for prolonged periods, leading to the accumulation of drug resistance mutations (DRMs) and undermining the effectiveness of both the current and future treatment options [3,5]. The absence of resistance testing in HIV treatment management can lead to premature switching to more costly and complex second-line regimens when first-line treatments could still be effective with improved adherence support. It may also result in the selection of suboptimal regimens that foster further resistance evolution, compromising the efficacy of second-line therapy and leaving patients with limited alternatives for long-term HIV management [4,6]. At the population level, insufficient monitoring can lead to the increased transmission of HIVDR strains, which complicates treatment strategies for newly infected individuals and potentially undermines the overall effectiveness of ART programs [5,6].

Ethiopia is one of the Sub-Saharan African countries most affected by the HIV-1 epidemic. In Ethiopia, ART is delivered through a public health approach that utilizes standardized first- and second-line treatment regimens along with simplified laboratory monitoring, which includes at least one viral load test annually. At the time of study enrollment, Ethiopia’s national ART guidelines recommended first-line regimens consisting of two nucleoside reverse transcriptase inhibitors (NRTIs)—tenofovir (TDF) with lamivudine (3TC)—combined with a non-nucleoside reverse transcriptase inhibitor (NNRTI) such as efavirenz (EFV) and nevirapine (NVP). However, the guidelines have evolved over time to incorporate newer, more effective drug combinations. Notably, dolutegravir (DTG), an integrase strand transfer inhibitor (INSTI), has been adopted as a preferred option in first-line regimens [7]. For second-line regimens, the guidelines advised the use of two NRTIs (such as zidovudine (ZDV) or abacavir (ABC) with 3TC) along with a ritonavir-boosted protease inhibitor, either lopinavir/ritonavir (LPV/r) or atazanavir/ritonavir (ATV/r) [7].

At the initiation of this study in 2017, Ethiopia had approximately 414,854 adults receiving ART out of an estimated 610,000 adults living with HIV, according to data from the Ethiopian Federal Ministry of Health. Of the patients receiving ART, approximately 6552 were on protease inhibitor/ritonavir (PI/r)-based second-line regimens [7,8]. Despite significant progress in scaling up ART in Ethiopia, routine virological monitoring and HIVDR testing are not yet standard practices in HIV care. As the number of individuals initiating first-line ART continues to grow, there is an anticipated increase in HIVDR and treatment failure, leading to a significant number of patients transitioning to second-line regimens [3,4]. Patients on second-line ART represent a unique population with histories of ART failure, longer durations of HIV infection, and a higher potential for accumulating HIVDRMs [9]. However, there is a notable lack of data on long-term outcomes and HIVDR patterns among patients on second-line regimens in Ethiopia. This study aims to assess the rate of virological failure (VF), the prevalence of HIVDR, and identify the factors associated with VF among patients on second-line ART in Ethiopia.

## 2. Methods

### 2.1. Study Design

In 2017, a cross-sectional study was conducted as part of a national survey to assess the prevalence of HIVDR in Ethiopia. This study included 28 ART sites, selected using probability proportional to size (PPS) sampling to ensure representation across various geographic regions and clinic sizes [10]. After selecting the clinics, we included all consecutive adults (≥18 years) who had been receiving second-line ART for at least six months during the study period based on a predetermined sample size that factored in the proportion of patients on second-line regimens at each facility. Written informed consent was obtained from all participants. Blood samples (10 mL) were collected via venipuncture from each participant for CD4+ T-cell counts, viral load measurements, and HIVDR genotyping. Basic demographic and clinical information were also collected using a standardized questionnaire. Plasma samples were transported on dry ice to the Ethiopian Public Health Institute (EPHI) for viral load testing and long-term storage at −80 °C. VL was quantified using the Abbott RealTime HIV-1 assay (Abbott Molecular Inc., Des Plaines, IL, USA). Samples with VL ≥ 1000 copies/mL were shipped to the National Institute of Respiratory Diseases (INER) laboratory in Mexico for HIVDR testing.

### 2.2. HIV-1 Sequencing

The HIVDR testing was performed at a WHO-designated laboratory in Mexico City, according to the WHO/HIV ResNet Laboratory Operational Framework [11]. Briefly, HIV RNA was extracted from 1 mL of plasma using the QIAamp Viral RNA Kit (QIAGEN, Valencia, CA, USA). Subsequently, the protease-reverse transcriptase (PR-RT) regions were amplified using validated in-house protocols [12]. The purified PCR fragments were then sequenced using the BigDye technology on the ABI Prism 3730 Genetic Analyzer (Applied Biosystems, Foster City, CA, USA). Sequence assembly and editing were performed using the RECallV 2.0 HIV-1 sequencing analysis tool (University of British Columbia, Vancouver, BC, Canada) [13]. Sequence quality control was performed using the WHO tool https://sequenceqc-dev.bccfe.ca/who_qc (accessed on 16 March 2024) and the Quality Control program of the Los Alamos HIV sequence database https://www.hiv.lanl.gov/ (accessed on 16 March 2024).

### 2.3. HIV-1 Drug Resistance Analysis

The HIVDR analysis was conducted using the Stanford HIV Drug Resistance Database (HIVdB v9.1) (https://hivdb.stanford.edu/hivdb/by-mutations, accessed on 29 June 2023). The prevalence of mutation patterns and their frequencies were calculated for each of the following drug classes: NRTIs, NNRTIs, and PIs. Additionally, an HIVDR susceptibility analysis was performed using the HIVdb algorithm, which assigns a drug penalty score for each antiretroviral drug. The cumulative scores were then used to classify the level of resistance into four categories: susceptible (score < 15), low-level resistance (score 15–29), intermediate resistance (score 30–59), or high-level resistance (score ≥ 60).

### 2.4. HIV-1 Subtype Determination

The HIV-1 subtyping was carried out using the online automated subtyping tools REGA v3.0 [14], COMET [15], and the jumping profile Hidden Markov Model (jpHMM) [16,17]. Subtyping was further validated by maximum likelihood (ML) phylogenetic tree analysis with the reference sequences from HIV-1 subtypes (A–K) and recombinant viruses, which were downloaded from the Los Alamos database (http://www.hiv.lanl.gov, accessed on 3 June 2023). Multiple sequence alignment was performed using MAFFT version 7 [18] and then manually edited using BioEdit V7.0.9.0 to achieve a perfect codon alignment [19,20]. The ML tree topology was constructed using the online version of PhyML v3.0 [21] with the GTR + I + Γ nucleotide-substitution model (utilizing the estimated proportion of invariable sites and four gamma categories) [16]. A heuristic tree search was conducted using the SPR branch-swapping algorithm. Branch support was evaluated with the approximate likelihood ratio test (aLRT-SH, Shimodaira Hasegawa-like) [22]. Clusters were defined as monophyletic clades with aLRT-SH support ≥ 0.9 [16]. The subtype-resolved ML phylogeny trees were visualized using the FigTree v1.4.0 program [23]. Sequences forming clusters with the reference sequences of the same subtype were assigned to that subtype.

### 2.5. Data Quality Management

Data quality assurance was rigorously maintained through a multi-step process. Every tenth entry of the retrospectively collected data was cross-checked, and all case and laboratory report forms were verified against the source data to ensure accuracy. Following this initial verification, the data were independently entered into Epi-Info V6.04 by two separate data clerks. This software was equipped with check programs to prevent the entry of erroneous data, adding an additional layer of quality control. Furthermore, manual consistency checks were performed to further enhance data integrity. Once these quality control measures were completed, the data were exported to SPSS v.26 for comprehensive analysis.

### 2.6. Statistical Analysis

Descriptive statistics were used to summarize demographic and clinical characteristics. The prevalence of VF (defined as having a viral load ≥ 1000 copies/mL) and acquired HIVDR was calculated, and associations between clinical characteristics and virological outcomes were assessed using logistic regression models. Different patient-related characteristics, including socio-demographic, clinical, immunological, and virological characteristics, were included in the analysis. Initially, variables were examined individually, and those demonstrating an association with VF at a *p*-value of < 0.2 were included in a multivariable logistic regression model. Prior to conducting the multivariable analysis, correlation analysis was performed to evaluate potential multicollinearity among variables, and no multicollinearity was detected. In the final multivariable model, predictors with *p*-values < 0.05 were considered statistically significant. Crude and adjusted odds ratios (ORs) with 95% confidence intervals (CIs) were reported to quantify the strength of the associations.

### 2.7. Ethical Considerations

The study protocol received approval from the Scientific and Ethical Research Office of the Ethiopian Public Health Institute (EPHI) and the Institutional Review Board of the Ethiopian Ministry of Science and Technology.

## 3. Results

### 3.1. Demographic Characteristics of Study Participants

A total of 586 participants were included in this study. Females comprised a greater portion of the sample at 54.44%, compared to males at 45.56%. The largest age group was individuals aged between 35 and 44 years (41.81%), followed by those aged 25 to 34 years (27.30%), over 45 years (24.91%), and 18 to 24 years (5.97%). Most participants resided in urban areas (84.47%), while 15.53% lived in rural regions. Regarding marital status, 39.93% were in monogamous marriages, 21.84% were unmarried, 19.80% were divorced, 9.39% were widowed, and 9.04% were in polygamous marriages. Educational attainment varied as follows: 44.20% had primary education, 23.21% had secondary education, 12.63% had college or university education, and 19.97% had no formal education. Occupational statuses were diverse, with housewives constituting 18.09%, government employees 17.06%, merchants 16.38%, unemployed individuals 15.36%, and students 4.27%. Other occupations, including drivers and various other jobs, accounted for 24.74% of the participants (Table 1).

### 3.2. ART Regimen and Clinical Characteristics of Patients on Second-Line Therapy

At the initiation of first-line ART, the most frequently prescribed regimen was AZT + 3TC + NVP, accounting for 26.79% of patients. This was followed by TDF + 3TC + EFV at 18.09%. Other combinations included D4T + 3TC + NVP (15.19%) and TDF + 3TC + NVP (14.33%). Less commonly used regimens were AZT + 3TC + EFV (12.46%), D4T + 3TC + EFV (5.12%), ABC + 3TC + NVP (6.14%), and ABC + 3TC + EFV (1.88%). The majority of patients (69.97%) switched after more than 91 months on first-line ARVs, while smaller proportions switched earlier as follows: 1.71% before 30 months, 7.17% between 31 and 60 months, and 21.16% between 61 and 90 months.

The most commonly prescribed second-line regimens were TDF + 3TC + ATV/r, used by 44.20% of participants, followed by ABC + 3TC + ATV/r (19.45%) and AZT + 3TC + ATV/r (17.58%). Smaller proportions were on TDF + 3TC + LPV/r (9.73%), AZT + 3TC + LPV/r (5.63%), and ABC + 3TC + LPV/r (3.41%). The duration of therapy on the second-line regimen varied as follows: the majority (55.80%) had been on second-line therapy for 16–48 months, 29.69% for 6–15 months, and 14.51% for over 49 months. At the initiation of ART, all participants had a CD4 count below 200 cells/mm^3^. By the time they switched to second-line treatment, 70.30% still had a CD4 count below 200 cells/mm^3^. However, at the time of this study, 71.00% of participants had achieved CD4 counts above 200 cells/mm^3^, while 29.00% still had CD4 counts below this threshold. Viral suppression, defined as an RNA viral load below 1000 copies/mL, was observed in 86.18% of participants, as shown in Table 2.

### 3.3. HIV Drug Resistance Mutation Profile

Among the 81 (13.82%) participants experiencing VF, genotyping was successfully performed for 37 individuals. Of these, 25 participants (67.56%) harbored at least one DRM. Dual-class DRMs, affecting both NRTI and NNRTI and/or PI classes, were observed in 18 participants (48.64%), while triple-class DRMs (involving PI, NRTI, and NNRTI) were identified in 7 participants (18.92%) (see Figure 1).

NRTI-associated DRMs were detected in 18 participants (48.64%), with M184V being the most frequent NRTI mutation, occurring in 22.86% of patients. Thymidine analogue mutations (TAMs) were detected in 38.57% of patients, with specific mutations, including T215Y/V (10.00%), K70E (8.57%), D67N (8.57%), and K219E (7.14%), while K65R and Y115F were found in 7.14% and 8.57%, respectively. NNRTI-associated DRMs were detected in 18 patients (48.64%). The most predominant NNRTI-associated mutations detected included K103N (18.87%), Y181C (13.21%), and G190A (13.21%). Additional mutations such as H221Y (9.43%), V106I/M (5.66%), V108I (5.66%), A98G (7.55%), and L100I (5.66%) were also detected. PI-associated DRMs were identified in 7 patients (18.92%), with the V82A mutation being the most prevalent, observed in 29.41% of cases. The I54V and M46I mutations were each detected in 23.53% of participants, while I50L (11.76%), L90M (5.88%), and N88S (5.88%) mutations were also observed (Table 3).

### 3.4. Factors Associated with Virological Failure

Although various factors were examined, occupation, duration on second-line ART, and current CD4 count were identified as factors associated with second-line ART VF. Among occupational groups, students demonstrated a markedly higher likelihood of experiencing VF, with an adjusted odds ratio (AOR) of 6.14 (95% CI: 1.93–19.48) compared to government employees, the reference group. Other occupations, including drivers (AOR = 2.10, 95% CI: 0.61–7.20), housewives (AOR = 1.02, 95% CI: 0.33–3.19), unemployed individuals (AOR = 1.67, 95% CI: 0.65–4.31), and those in the “other” category (AOR = 1.57, 95% CI: 0.67–3.68), did not show statistically significant associations. The duration of second-line ART was another critical factor associated with VF. Participants who had been on second-line ART for over 12 months had a significantly lower likelihood of virological failure (AOR = 0.52, 95% CI: 0.28–0.97) compared to those on ART for less than 12 months. Additionally, current CD4 count emerged as a significant determinant of outcomes. Participants with a current CD4 count of less than 200 cells/mm³ were more likely to experience VF (AOR = 2.55, 95% CI: 1.52–4.28) compared to those with a CD4 count at or above 200 cells/mm^3^ (Table 4).

### 3.5. HIV Drug Resistance Susceptibility Testing

Our antiretroviral drug resistance analysis reveals varying degrees of susceptibility across the three primary ARV classes (NRTIs, NNRTIs, and PIs). Among NRTIs, drugs like AZT and TDF showed higher susceptibility rates at 78.38% and 62.16%, respectively, while drugs like emtricitabine (FTC) and 3TC had significant resistance, with 43.24% exhibiting high levels of resistance. Drugs like D4T and DDI had 59.46% susceptibility, with a notable portion (24.32% and 27.03%, respectively) having high resistance. Among NNRTIs, DOR and ETR had 54.05% susceptibility, with high-level resistance in 8.11% of samples. EFV and NVP showed lower susceptibility rates of 35.14%, accompanied by high-level resistance in 48.65% and 59.46% of samples, respectively. RPV exhibited 51.35% susceptibility, while high-level resistance was observed in 37.84% of samples. PIs generally maintained high levels of susceptibility. DRV was fully susceptible (100%), while ATV demonstrated susceptibility in 81.08% of samples and high-level resistance in 16.22%. FPV, IDV, LPV, NFV, and SQV exhibited susceptibility rates of 83.78% to 86.49%, with high-level resistance detected in 5.41% to 13.51% of samples. TPV showed 86.49% susceptibility, with no case of high-level resistance (Table 5).

### 3.6. Maximum-likelihood Phylogenetic Tree

The phylogenetic tree in Figure 2 comprises 459 sequences, including 37 Ethiopian sequences and 422 reference sequences for HIV-1 subtypes (A–K) and circulating recombinant forms, which were downloaded from the HIV-1 LANL database. An ML tree was constructed using the online version of PhyML v 3.0. In the figure, the reference sequences from the Los Alamos National Laboratory are shown in black. Ethiopian sequences clustering with the HIV-1 subtype C reference sequence are depicted in blue, while non-subtype C Ethiopian sequences are shown in pink. Phylogenetic analysis of the sequences showed that 94.59% (35 out of 37) clustered with HIV-1 subtype C. The remaining two sequences, accounting for 2.70% each, were identified as non-subtype C, specifically subtype B, and a recombinant form of subtypes D and A1 (see Figure 2).

## 4. Discussions

This study provides the first national data on viral suppression and HIVDR among patients on second-line ART in selected health facilities in Ethiopia. Our findings showed that 13.82% of patients on second-line ART experienced VF, defined as a VL ≥ 1000 copies/mL. Notably, 32.43% of study participants did not harbor any DRMs, suggesting that poor adherence to treatment may be the primary cause of VF in these cases. Among patients with VF, 67.57% exhibited at least one class of DRMs, with 48.64% showing dual-class DRMs and 18.921% displaying triple-class DRMs. The low PI-associated DRMs (18.92%) and high susceptibility to drugs like darunavir, atazanavir, and lopinavir suggest that PI-based regimens remain an effective backbone for second-line treatments in Ethiopia if adherence issues are addressed. We also identified risk factors for VF, including being a student (AOR = 6.14; 95% CI: 1.93–19.48), shorter duration on second-line treatment (less than 12 months, AOR = 0.52; 95% CI: 0.28–0.97), and a CD4 count below 200 cells/mm³ (AOR = 2.55, 95% CI: 1.52–4.28).

Our study showed that 13.82% of patients on second-line antiretroviral therapy (ART) experienced VF (viral load ≥1000 copies/mL). This finding is consistent with VF rates reported in several other African countries, including Tanzania, Rwanda, Uganda, South Africa, and Ethiopia, where rates ranged from 12% to 41% [24,25,26,27,28,29,30,31]. A systematic review and meta-analysis focusing on patients undergoing second-line ART regimens in resource-limited settings further corroborated our results, demonstrating cumulative VF rates between 21.8% and 38.0% [25]. Our findings, along with those from other studies, underscore the persistent challenge of maintaining viral suppression in patients on second-line therapy across various resource-limited settings.

Among our study participants, 32.4% did not harbor any DRMs, suggesting that poor adherence to treatment may be the primary cause of VF. This is consistent with previous research in resource-limited settings, which revealed that 33–67% of patients on second-line PI-based regimens with VF had no major DRMs, suggesting adherence issues rather than drug resistance as the primary cause of failure [10,24,29,32,33]. For example, among patients with VF, 37% in Rwanda [29], 58.5% in Namibia [10], and 33% in South Africa [32] did not harbor any DRMs. Identified risk factors for second-line failure include suboptimal adherence, prior exposure to non-standard regimens, and pre-existing PI resistance [34,35]. However, it also raises concerns that some patients may have been prematurely switched to second-line regimens without resistance testing, underscoring the need for resistance testing to ensure optimal treatment decisions.

Among patients experiencing VF, a significant proportion demonstrated complex resistance patterns. Dual-class DRMs affecting both NRTI and NNRTI classes and/or protease inhibitors were identified in 48.64% of patients. Even more concerning was the presence of triple-class DRMs in 18.92% of cases, involving mutations across all the following three major drug classes: PIs, NRTIs, and NNRTIs. These results are consistent with previous studies in resource-limited settings, where dual-class resistance in patients on second-line ART was found in 40% to 60% of cases [29,36]. Furthermore, the observed rate of triple-class resistance (18.92%) in this cohort closely aligns with findings from other studies in sub-Saharan Africa, where 22% of participants experiencing failure on second-line therapy in routine care settings also showed triple-class resistance [36]. However, this rate is lower than that reported in a study from India, which showed that 53% of patients harbored triple-class resistance [36]. The findings from this study, along with evidence from prior research, highlight a persistent and widespread issue of HIVDR among patients on second-line ART across various resource-constrained environments. The high prevalence of dual- and triple-class DRMs observed highlights the significant challenges in managing second-line ART failures in these settings. Such multi-class resistance substantially limits future treatment options, potentially necessitating the use of third-line regimens that are typically more expensive and less readily available. These results emphasize the critical need for enhanced viral load monitoring, resistance testing, and improved access to third-line ART in Ethiopia and similar resource-limited contexts [37,38]. Additionally, our findings underscore the importance of assessing and supporting medication adherence, as poor adherence remains a major contributor to VF [29].

We observed a 48.64% prevalence of NRTI-associated DRMs, with the most common being M184V and TAMs, present in 22.86% and 38.57% of patients, respectively. M184V is known to emerge rapidly in patients receiving 3TC or FTC, which are common components of first-line regimens [39,40]. The prevalence of TAMs (T215Y/V, K70E, D67N, and K219E) is particularly concerning, as these mutations accumulate over time and can confer cross-resistance to multiple NRTIs, significantly impacting the effectiveness of NRTI-based regimens [39]. Additionally, the K65R mutation, detected in 7.14% of patients, raises additional concern due to its association with broad resistance, especially to TDF and ABC, which are critical components of second-line regimens in resource-limited settings like Ethiopia. Similar resistance patterns have been documented in studies conducted among patients on second-line ART in various African countries, including Rwanda, Uganda, Namibia, Gabon, South Africa, and Zambia [29,41,42,43,44]. These complex mutation patterns, characterized by high rates of M184V mutations and TAMs, are typically attributed to several common factors in these settings as follows: limited availability of resistance testing, prolonged periods of VF while on first-line therapy, the use of suboptimal treatment regimens, and challenges with medication adherence [45,46]. These circumstances collectively contribute to the development and accumulation of DRMs, potentially compromising the efficacy of second-line ART regimens and complicating long-term HIV management in these regions.

We detected NNRTI-associated DRMs in 65% of patients, even though NNRTIs are typically excluded from second-line ART regimens. This indicates persistence of NNRTI-resistant viral strains even after discontinuation of NNRTI-based therapies [39]. Similar results from other studies have shown that NNRTI-associated DRMs often remain detectable in patients transitioning to second-line regimens following failure of first-line treatments [43,44,45,47]. The most frequently observed NNRTI-associated mutations include K103N, Y181C, and G190A. These mutations are known to confer high-level resistance to first-generation NNRTIs such as efavirenz and nevirapine, which are commonly used in first-line regimens [46]. Furthermore, the additional mutations detected, such as H221Y, V106M, V108I, A98G, and L100I, can contribute to cross-resistance within the NNRTI class and may impact the effectiveness of newer NNRTIs like etravirine and rilpivirine [48]. Our findings are consistent with findings from several studies conducted in resource-limited settings [44,49,50,51].

Our study’s finding of an 18.92% prevalence of PI-associated DRMs among patients on second-line regimens is supported by previous research in similar settings [44,45]. A pooled analysis of PI-based treatment failures in sub-Saharan Africa reported that 17% of patients had at least one major PI-resistance mutation at the time of treatment failure [43]. Studies from Namibia, Uganda, and South Africa reported 13.1%, 19.4%, and 22% of PI-associated HIVDRMs, respectively, among people receiving PI/r-based second-line ART [29,41,44,52]. However, our results are lower than those reported in other studies. For instance, Chimukangara et al. found in South Africa that among 348 samples analyzed, 287 (82.5%) had at least one DRM and 114 (32.8%) had at least one major PI-resistance DRM [53]. Similarly, a systematic review conducted in Asia by Ross et al. revealed that 13 out of 39 patients (33%) had major PI DRMs [54].

The V82A mutation, detected in 29.41% of patients, is a major PI resistance mutation that can significantly reduce susceptibility to several PIs, including lopinavir and indinavir. The I54IV/V and M46I mutations, each found in 23.53% of patients, are also major PI resistance mutations that can contribute to reduced PI efficacy. The presence of I50L, L90M, and N88S mutations, albeit at lower frequencies, further underscores the complexity of PI resistance patterns in this cohort. The I50L mutation is particularly associated with resistance to atazanavir, while L90M can confer broad cross-resistance to multiple PIs [25,36]. Several studies among patients receiving second-line PI-based regimens have documented similar PI-associated mutations [42,43,44,45].

The relatively low prevalence of PI mutations (18.92%) and the high level of susceptibility to DRV, ATV, and LPV among the majority of participants in our study suggest that PI-based regimens may still be an effective backbone of most second-line regimens for many patients failing second-line therapy, provided adherence issues are addressed. However, the presence of PI mutations in some patients underscores the need for careful monitoring and resistance testing to ensure these patients are prescribed effective third-line ART regimens.

Our result highlights a significant association between the duration of second-line ART and viral load suppression. Participants who had been on second-line ART for over 12 months had significantly lower odds of viral load failure compared to those treated for less than 12 months (AOR: 0.52, 95% CI: 1.52–4.28). The improved outcomes associated with longer ART duration may be attributed to enhanced medication adherence over time [55]. As patients become more accustomed to their treatment regimens and learn to manage side effects, adherence likely improves, resulting in better viral suppression. Additionally, extended periods of viral suppression reduce the risk of DRMs, which are more likely to arise when viral replication is poorly controlled. This is especially important in the context of second-line therapy, where treatment options become more limited, and maintaining the effectiveness of available drugs is crucial. Early identification of at-risk individuals through viral load monitoring and adherence support can improve long-term outcomes. Consistent with previous research, our findings suggest that providing intensive adherence support and regular VL monitoring during the early stages of second-line therapy is critical to enhance the likelihood of achieving and maintaining viral suppression and treatment success over time [45,53]. This is more significant in resource-limited settings where third-line ART options are limited.

Our study also reveals significant associations between occupation and VF among participants on second-line ART. Students emerged as the group with the highest risk (AOR: 6.14, 95% CI: 1.93–19.48) compared to government employees. Students may face unique adherence difficulties, such as balancing academic responsibilities, social life, and health management [56]. Furthermore, as young adults, they may experience greater stigma or lack of access to consistent healthcare, making adherence more challenging [56]. Additionally, stress, depression, and anxiety are prevalent among students and can disrupt consistent ART adherence. Our findings underscore the importance of designing interventions tailored to students’ unique circumstances, including flexible healthcare delivery models (e.g., mobile health units and telemedicine), enhanced adherence support, and psychosocial counseling to address their specific challenges [57].

Although not statistically significant, individuals in occupations such as drivers, unemployed persons, and those classified under “other” categories demonstrated increased odds of VF. For example, drivers often contend with irregular work schedules and extended periods away from home, which can disrupt medication adherence and hinder regular clinic attendance. To address these challenges, adherence interventions for drivers could involve strategies such as providing multi-month medication supplies or offering flexible clinic appointment times. These measures would help ensure that drivers can maintain their treatment regimens despite the demands and unpredictability of their work environments [58]. Overall, these findings highlight the critical importance of ensuring comprehensive adherence support for all patients on second-line ART, regardless of their occupation.

Our results showed a relationship between CD4 count and the likelihood of viral load failure among individuals on second-line ART. Specifically, participants with a CD4 count below 200 cells/mm^3^ were found to have 2.55 times higher odds (AOR: 2.55; 95% CI: 1.52–4.28) of experiencing VF compared to those with CD4 counts at and above this threshold. This suggests that individuals with lower CD4 counts, indicative of more pronounced immunosuppression, are at heightened risk of treatment failure. Our findings align with other studies that similarly report an increased risk of VF associated with low baseline CD4 counts, which is often accompanied by HIVDR [27,28,30,59,60].

The genetic diversity of HIV-1 in the study group aligns with the typical diversity observed in different studies in Ethiopia. It highlights the clear dominance of subtype C in the country, with 94.59% (35 out of 37) of the sequences clustering within this subtype [17,61,62,63].

### Limitation of This Study

This study marks a significant milestone as the first national survey to encompass multiple health facilities, yet it is not without its limitations. The relatively small sample size poses a challenge, potentially failing to accurately represent the broader population of patients on second-line therapy, which in turn restricts the generalizability of the findings. We defined VF as a single viral load measurement of ≥1000 copies/mL, without the requirement for a second confirmatory test. This approach could potentially overestimate the incidence of VF. The overall genotyping success rate was 45.68%, which might have influenced the study results. Moreover, the cross-sectional nature of the study design only provides a snapshot of resistance mutations at a single point in time, preventing any assessment of longitudinal changes or the progression of these mutations over an extended period. To address these limitations and gain a more comprehensive understanding of the evolution of resistance mutations and their clinical implications among patients on second-line regimens, future research should focus on incorporating larger, more diverse cohorts and implementing longitudinal follow-up strategies.

## 5. Conclusions

This study, the first national HIVDR survey conducted in selected health facilities in Ethiopia, indicates that 13.82% of patients on second-line ART did not achieve viral suppression. This study identified both dual-class and triple-class resistance among those experiencing virological failure, indicating a substantial challenge in managing HIVDR. This underscores the need for comprehensive HIV care in Ethiopia, which should include adherence support, regular viral load monitoring, and HIVDR testing for patients on second-line ART. Notably, the relatively low prevalence of PIs mutations and the high level of susceptibility to PIs suggest that PI-based regimens could remain effective as the backbone of second-line regimens, provided that adherence issues are adequately addressed.

## Figures and Tables

**Figure 1 viruses-17-00206-f001:**
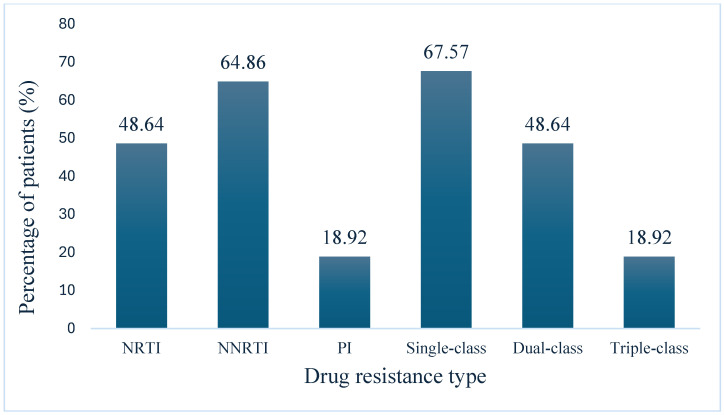
The proportion of patients with HIV drug resistance mutation.

**Figure 2 viruses-17-00206-f002:**
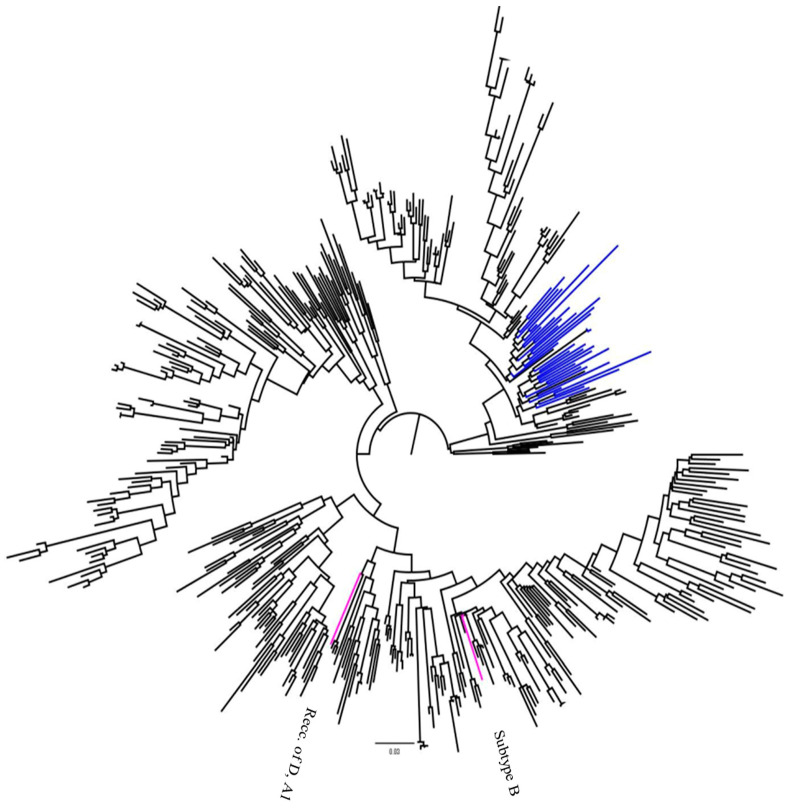
The maximum-likelihood phylogenetics tree of HIV-1 subtypes and circulating recombinant forms circulating in Ethiopia. Subtype C reference sequences are depicted in blue, while non-subtype C Ethiopian sequences are shown in pink.

**Table 1 viruses-17-00206-t001:** Demographic characteristics of patients on second-line treatment in Ethiopia (*n* = 586).

Characteristics	*N* (*n* = )	Percent (%)
Age (Year)		
18–24	35	5.97
25–34	160	27.30
35–44	245	41.81
≥45	146	24.91
Sex		
Male	267	45.56
Female	319	54.44
Residence		
Urban	495	84.47
Rural	91	15.53
Marital status		
Unmarried	128	21.84
Married, monogamous	234	39.93
Married, polygamous	53	9.04
Divorced	116	19.80
Widowed	55	9.39
Educational level		
No formal education	117	19.97
Primary education	259	44.20
Secondary education	136	23.21
College or university	74	12.63
Occupational status		
Government employee	100	17.06
Merchant	96	16.38
Driver	24	4.10
Housewife	106	18.09
Student	25	4.27
Unemployed	90	15.36
Other	145	24.74

**Table 2 viruses-17-00206-t002:** Clinical characteristics and laboratory parameters of participants with HIV on second-line treatment.

Characteristics	*N* (*n* = )	Percent (%)
Second-line ARV regimens	*n* = 586	
ABC + 3TC + ATV/r	114	19.45
ABC + 3TC + LPV/r	20	3.41
AZT + 3TC + ATV/r	103	17.58
AZT + 3TC + LPV/r	33	5.63
TDF + 3TC + ATV/r	259	44.20
TDF + 3TC + LPV/r	57	9.73
Duration on second-line regimen (months)	*n* = 586	
6–15	174	29.69
16–48	327	55.80
≥49	85	14.51
First-line ARV regimens	*n* = 586	
ABC + 3TC + EFV	11	1.88
ABC + 3TC + NVP	36	6.14
AZT + 3TC + EFV	73	12.46
AZT + 3TC + NVP	157	26.79
D4T + 3TC + EFV	30	5.12
D4T + 3TC + NVP	89	15.19
TDF + 3TC + EFV	106	18.09
TDF + 3TC + NVP	84	14.33
Duration on first-line regimens (months)	*n* = 586	
0–30	10	1.71
31–60	42	7.17
61–90	124	21.16
≥91	410	69.97
Baseline CD4 at ART initiation (cell/mm^3^)	*n* = 548	
<200	0	0
≥200	548	100
CD4 at second-line initiation (cell/mm^3^)	*n* = 541	
<200	541	100
≥200	0	0
Current CD4 (during study (cell/mm^3^)	*n* = 555	
<200	161	29.00
≥200	394	71.0
Viral load (RNA copies/mL)	*n* = 586	
<1000	505	86.18
≥1000	81	13.82

Abbreviations: antiretroviral (ARV), antiretroviral therapy (ART), abacavir (ABC), lamivudine (3TC), efavirenz (EFV), zidovudine (AZT), atazanavir/ritonavir (ATV/r), lopinavir/ritonavir (LPV/r), nevirapine (NVP), tenofovir (TDF), stavudine (D4T).

**Table 3 viruses-17-00206-t003:** Drug resistance mutation profiles among patients on second-line regimens with virological failure.

Mutation Type	*N* (*n* = )	Percent (%)
NRTIs		
A62V	2	2.86
D67N/G	6	8.57
E44ED	4	5.71
K65R	5	7.14
K70E/G/R/T	6	8.57
L210W	3	4.29
L74I	1	1.43
M184V	16	22.86
K219E	5	7.14
M41L	2	2.86
T215Y/V	7	10.00
T69SADN	2	2.86
Y115F	6	8.57
S68G	5	7.14
NNRTIs		
A98G	4	7.55
E138A	2	3.77
G190A/E	7	13.21
H221Y	5	9.43
K101E/A	4	7.55
K103N	10	18.87
K238T	1	1.89
L100I	3	5.66
P225H	1	1.89
V106I/M	3	5.66
V108I	3	5.66
V179T/D	2	3.77
Y181C	7	13.21
Y188L	1	1.89
PIs		
I50L	2	11.76
I54V	4	23.53
L90M	1	5.88
M46I	4	23.53
N88S	1	5.88
V82A	5	29.41

**Table 4 viruses-17-00206-t004:** Multivariate analysis of factors associated with virological failure among patients on second-line ART.

Variables	COR (95% CI)	*p*-Value	AOR (95% CI)	*p*-Value
Sex				
Male	Ref.			0.07
Female	0.53 (0.33–0.85)	0.008	0.58 (0.32–1.05)	
Occupation				
Government employee	Ref.			
Merchant	0.95 (0.40–2.30)	0.906	1.15 (0.44–3.06)	0.772
Driver	1.93 (0.61–6.13)	0.265	2.10 (0.61–7.20)	0.243
Housewife	0.60 (0.23–1.53)	0.285	1.02 (0.33–3.19)	0.973
Student	4.13 (1.50–11.40)	0.006	6.14 (1.93–19.48)	0.002 *
Unemployed	1.35 (0.59–3.10)	0.478	1.67 (0.65–4.31)	0.29
Other	1.31 (0.62–2.80)	0.481	1.57 (0.67–3.68)	0.298
Duration on ART at second-line (months)				
<12	Ref.			
>12	0.53 (0.30–0.93)	0.028	0.52 (0.28–0.97)	0.039 *
Current CD4 (cell/mm^3^)				
<200	2.55 (1.54–4.21)	<0.001	2.55 (1.52–4.28)	<0.001 *
≥200	Ref.			

Abbreviations: AOR, adjusted odds ratios; COR, crude odds ratios; CI, confidence interval. (*) Statistically significant at *p* ≤ 0.05.

**Table 5 viruses-17-00206-t005:** HIV drug resistance susceptibility test among the study participants (*n* = 37).

Resistance Level (*n* (%))				
Antiretroviral Drug	Susceptible	Low Level of Resistance	Intermediate Level of Resistance	High Level of Resistance
NRTIs				
Abacavir (ABC)	21 (56.78%)	2 (5.41%)	2 (5.41%)	12 (32.43%)
Zidovudine (AZT)	29 (78.38%)	1 (2.70%)	0	7 (18.92%)
Stavudine (D4T)	22 (59.46%)	3 (8.11%)	3 (8.11%)	9 (24.32%)
Didanosine (DDI)	22 (59.46%)	2 (5.41%)	3 (8.11%)	10 (27.03%)
Emtricitabine (FTC)	21 (56.76%)	0	0	16 (43.24%)
Lamivudine (3TC)	21 (56.76%)	0	0	16 (43.24%)
Tenofovir (TDF)	23 (62.16%)	2 (5.41%)	6 (16.22%)	6 (16.22%)
NNRTIs				
Doravirine (DOR)	20 (54.05%)	6 (16.12%)	8 (21.62%)	3 (8.11%)
Efavirenz (EFV)	13 (35.14%)	2 (5.41%)	4 (10.81%)	18 (48.65%)
Etravirine (ETR)	20 (54.05%)	6 (16.21%)	8 (21.62%)	3 (8.11%)
Nevirapine (NVP)	13 (35.14%)	0	2 (5.41%)	22 (59.46%)
Rilpivirine (RPV)	19 (51.35%)	2 (5.41%)	2 (5.41%)	14 (37.84%)
PIs				
Atazanavir (ATV)	30 (81.08%)	1 (2.70%)	0	6 (16.22%)
Darunavir (DRV)	37 (100%)	0	0	0
Fosamprenavir (FPV)	32 (86.49%)	1 (2.70%)	2 (5.41%)	2 (5.41%)
Indinavir (IDV)	31 (83.78%)	1	1 (2.70%)	4 (10.81%)
Lopinavir (LPV)	31 (83.78%)	1 (2.70%)	0	5 (13.51%)
Nelfinavir (NFV)	31 (83.78%)	0	1 (2.70%)	5 (13.51%)
Saquinavir (SQV)	31 (83.78%)	1 (2.70%)	1 (2.70%)	4 (10.81%)
Tipranavir (TPV)	32 (86.49%)	3 (8.11%)	2 (5.41%)	0

## Data Availability

All the sequences from this study will be deposited in GenBank.

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
