# Peer review of "Viral Suppression and HIV Drug Resistance Among Patients on Second-Line Antiretroviral Therapy in Selected Health Facility in Ethiopia"

_viruses, 2025, doi:10.3390/v17020206_

Round 1
Reviewer 1 Report
Comments and Suggestions for Authors
The authors used a cross-sectional study to investigate the virological failure (VF), prevalence of HIV drug resistance and factors associated with VF among patients on second-line ART in Ethiopia. The information should be helpful for HIV treatment management in Ethiopia. I suggest the authors to clarify the study design and analysis method and better align the objective, method with the results. In addition, it might be helpful to clarify what can be added by this study to the existing literature since a similar study about factors associated with VF in Ethiopia was published (Zakaria, et al., 2022).
Line 25, Consider use alternative word instead of “significant” here since it might confuse the readers with statistical significance.
Line 81-82, Please clarify. Is the proportional-to-size sampling to ensure representation of clinics providing ART or HIV infected subjects across various geographic regions and clinic sizes? Also, consider providing the reference with more information on sampling or recruitment of the study design if it is available.
Line 83-85, Were the 586 participants included in this study all subjects with the second line treatments from the 28 ART sites or the participants were convenient samples from the 28 sites?
Line 143-145, Please specify how the outcome variable, virological failure (VF), was defined, what factors were assessed, and the approach used for factor selection in the logistic model. Is the analysis here referring to the logistic model? Were all factors listed in Table 1 and Table 2 assessed by the logistic regression model approach? Was a forward, backward, or stepwise selection elimination approach used here? Were the interaction terms checked?
Line 194, Please consider including the virological failure prevalence since it was stated in the method (line 143). Were all the other subjects reached viral suppression? Why are there only 37 out of 81 participants with the genotyping profiles?
Line 213-227, Consider using “factor associated with second-line ART VF” instead of “predictor” as in the Method section. (The model was assessed for the predictability.) In addition, the authors have specified in the method section that the significant level were p=0.05. The p_values should be included, or significant factors should be marked in Table 4.
Line 231, Were the second line regiments included in the logistic regression model to assess VF? If so, what might be the reason that those ARTs with high resistance level were not found to be significantly associated with VF in the logistic regression model?
Line 275, Were the factors identified in this study similar to the factors by the study conducted earlier in Ethiopia (Zakaria et al., 2022)?
Line 279-283, the prevalence of patients on second-line ART experienced VF (13.82%) is on the lower side compared to the AF rate found in the other resource-limited countries. What might be the reasons that Ethiopia is doing better than the other countries (i.e., Tanzania, Rwanda, Uganda, South Africa)?
Line 295-297, since the participants in this study were from the ART clinics, were the data about resistance testing available to see the number of participants on the second-line regimens had resistance test before switching?
Table 2: Baseline CDC 4 at initiation of ART, the percentage did not sum up to 1.
For Table 1-Table3, for column 2, consider using the label “N (n=)” replace “frequency” and specify the number.
Author Response
Response has been attached

Reviewer 2 Report
Comments and Suggestions for Authors
The reviewer had a positive impression after reading the manuscript. Overall, the reviewer has no critical comments on the work. At the same time, two questions arose while reading. And adding information about this to the manuscript, in the reviewer's opinion, will improve the manuscript.
1. Lines 194-195. It is indicated that sequencing was successfully performed for 37 patients out of 81 who experienced virological failure. This is 43.5%. The WHO guidelines indicate that a satisfactory level is considered to be successful sequencing of more than 80% of collected clinical samples with a sufficient concentration of HIV RNA. The level obtained in the work is quite low, so it is recommended to add a phrase describing the reasons for obtaining such results.
2. Line 195. It is indicated that among the 37 successfully sequenced patients, 25 (67.56%) had DR mutations. According to the literature, if the resistance test is prescribed correctly, mutations should be detected in more than 80% of patients. And if mutations are not detected in the case of virological failure, it is recommended to analyze the concentration of HIV RNA in the sample. If the concentration is high in the absence of DR mutations, then there is a possibility that the patient has not been taking ARV drugs for some time. Such cases usually indicate either insufficient qualifications of physicians treating HIV infection and prescribing the DR test, or their high workload, which does not allow them to devote sufficient time to analyzing the patient's medical history. Due to the fact that the detected level is 67.56%, it can be assumed that such cases could have occurred. And in this regard, it is recommended to conduct two additional analyses. First, compare the viral load in a group of patients (25 people) with a drug-resistant variant of the virus with a group of patients with HIV without DR mutations (12 people). It is possible that the difference in VL between these groups will be reliable. Secondly, since the study was multicenter, it is possible to compare the centers and estimate what proportion of sequenced patients in each center had drug-resistant virus. Higher values ​​will indicate better quality of medical care and more competent assignment of HIV drug resistance testing.
Also, it can be noted that a small number of minor errors were found in the work (line 58 - missing ")."; table 2 – M184V/V), which, however, do not impair the understanding of the work.
Author Response
Response has been attached

Round 2
Reviewer 1 Report
Comments and Suggestions for Authors
Thank you for the helpful responses. Based on your responses, the design of the survey used 2 stage probability proportional to size sampling strategy. Was the analysis conducted accounting for the sampling weights and clustering effect? Based on the design, to produce population estimates (e.g., national prevalence), the analysis needs to account for the survey weights and clustering effect. Interactions should also be assessed in the weighted logistic model unless it is not plausible.
Line 83: Please check. “Proportion to size” sampling should be “probability proportion to size” sampling.
Line 151. Is the “logistic regression” here “weighted logistic regression”?
